# Comparison of Two Techniques Performing the Supine-to-Sitting Postural Change in Patients with Sternotomy

**DOI:** 10.3390/jcm12144665

**Published:** 2023-07-13

**Authors:** Marica Giardini, Marco Guenzi, Ilaria Arcolin, Marco Godi, Massimo Pistono, Marco Caligari

**Affiliations:** 1Istituti Clinici Scientifici Maugeri IRCCS, Division of Physical Medicine and Rehabilitation of Veruno Institute, 28013 Gattico-Veruno, Italy; marica.giardini@icsmaugeri.it (M.G.); marco.godi@icsmaugeri.it (M.G.); 2Istituti Clinici Scientifici Maugeri IRCCS, Division of Cardiac Rehabilitation of Veruno Institute, 28103 Gattico-Veruno, Italy; marco.guenzi@icsmaugeri.it (M.G.); massimo.pistono@icsmaugeri.it (M.P.); 3Istituti Clinici Scientifici Maugeri IRCCS, Integrated Laboratory of Assistive Solutions and Translational Research (LISART), Scientific Institute of Pavia, 27100 Pavia, Italy; marco.caligari@icsmaugeri.it

**Keywords:** perceived effort, pain, postural change, rehabilitation, sternal instability, sternotomy, supine-to-sitting

## Abstract

Patients with sternotomy are advised to follow sternal precautions to avoid the risk of sternal complications. However, there are no standard recommendations, in particular to perform the supine-to-sitting postural change, where sternal asymmetrical force may be applied. The aim of this study was to compare the rotational movement and the use of a tied rope (individual device for supine-to-sitting, “IDSS”) to perform the supine-to-sitting postural change. A total of 92 patients (26% female) admitted to a rehabilitative post-surgery ward with sternotomy were assessed for sternal instability. Levels of pain and perceived effort during the two modalities of postural change and at rest were assessed. Patients reported higher values of pain and perceived effort (both *p* < 0.0005) during rotational movement with respect to the use of the IDSS. Moreover, patients with sternal instability (14%) and female patients with macromastia (25%) reported higher pain than those stable or without macromastia (both *p* < 0.05). No other risk factors were associated with pain. Thus, the IDSS seems to reduce the levels of pain and perceived effort during the supine-to-sitting postural change. Future studies with quantitative assessments are required to suggest the adoption of this technique, mostly in patients with high levels of pain or with sternal instability.

## 1. Introduction

Median sternotomy is the gold standard in cardiac surgery incisions for the treatment of all congenital and acquired heart diseases [1]. It is performed on about one million patients every year all over the world [2]. However, sternotomy is not immune from complications, such as dehiscence, mediastinitis, osteomyelitis, and sternal displacements [3]. Sternal displacements are the most common sternal complication following cardiac surgery [4]. This is due to micromotion between the two halves of the wired sternum [5]. Sternal displacements have an underestimated incidence of about 1–8% [6,7,8], which causes an increased mortality of 10–40% in patients. It should therefore be clinical practice to prompt the patient to avoid sudden movements during the acute phase, which may result in high traction forces to the sternal edges [9]. Additionally, repetitive inadequate movements can cause a gradual gliding trauma in the post-acute phase. This can develop into a sternal separation produced mainly by lateral distraction forces, usually accompanied by pain syndromes [10].

Cough and asymmetrical movements with upper limbs and trunk are the main risk factors for sternal displacement [4,11,12,13,14]. Thus, optimising timely intervention and providing the patient with detailed indications on how to perform daily life activities might reduce pain, morbidity, mortality, and total cost of care [1,2,15,16,17,18]. These indications are called “sternal precautions” [19]. However, in recent years, some studies agreed that this term is excessively restrictive. It was suggested to adopt pain and/or sternum discomfort as a marker to identify the possible presence of sternal instability; in this view, patients should be instructed to perform daily life activities within a pain-free range [17,20]. Recent studies showed that providing patients with instructions to encourage the use of upper limbs within pain limit or discomfort during daily activities did not increase the risk of sternal complications with respect to conventional “restrictive” precautions [17,21]. Moreover, it did seem that restrictive guidelines could induce self-efficacy in patients, promoting anxiety and depression [22].

Although the health facilities affect how indications are provided, 90% of the physiotherapists offered pre- and post-operative information, such as early mobilisation, post-sternotomy restrictions, techniques for bed/chair mobility, breathing exercises, coughing techniques, and information about exercising the lower extremities [23,24]. While exercises, such as cardiopulmonary and general mobility, are usually performed for a few minutes a day, and with the assistance or supervision of the physiotherapist, postural changes are the most common movements, repeated several times in the day. The postural change that particularly urges with shear forces to the sternum is the “supine-to-sit” [24]. The two most common methods to perform this postural change are (1) the “rotational” one (with the indication: “Keep your move into the tube” by Adams et al. [25]), i.e., passage from the supine position to the lateral decubitus position and then, pushing with the arms on the bed, to the sitting position [26]; and (2) the use of a rope (which will be called individual device for supine-to-sitting “IDSS”) tied to the bottom of the bed, to be pulled symmetrically with the upper limbs in order to reach the sitting position. Although the second modality is quite usual in post-operative settings in improving autonomy and in reducing the risks, there is lack of evidence in literature. Up to now, no studies investigated implications for the sternal wound, and apparently no comparison study between the two methods exists.

Thus, the aim of the present study was to evaluate which of the two postural change modalities, through a rotational movement or performed using the IDSS, was less painful and evoked less perceived effort, in order to identify which modality should be recommended to patients with sternotomy.

## 2. Materials and Methods

### 2.1. Participants

This observational study was conducted between April and October 2022, at the Istituti Clinici Scientifici Maugeri IRCCS, Institute of Veruno, Italy. Patients were recruited among those admitted to the Cardiac Rehabilitation ward, those of which underwent surgery with median sternotomy under general anesthesia. All subjects met the following inclusion criteria: (1) presence of pain due to the sternotomy, defined as a score of at least 1 point on the numeric rating scale (NRS), during the postural change supine-to-sitting; (2) normal cognitive functioning (maximum of 0–2 errors in the Short Portable Mental State Questionary [27]); (3) ability to provide informed consent; (4) ability to get out of bed autonomously; and (5) stable clinical parameters. Exclusion criteria were other important comorbidities (i.e., orthopedic surgery < 6 months before, neurological disease, etc.), the presence of medical devices that limit movement (i.e., drainage) or the medical prescription of wearing sternal support harness.

The study protocol was approved by the Ethics Committee of the Istituti Clinici Scientifici Maugeri, Italy (approval number #2448 CE) and all participants signed the informed consent.

### 2.2. Assessment and Data Collection

All subjects were evaluated the day of admission by a physiotherapist with at least ten years of experience in cardiac rehabilitation. Age, sex, height, weight, smoking history, comorbidities, presence of macromastia in women (EU bra size > 80), and measurement of chest circumference in an upright position, at the mamillar line with the use of a tape measure, were recorded. The following evaluations were used:

#### 2.2.1. Sternal Instability Scale (SIS)

SIS is a 4-point scale that aims to assess the stability of the sternum and assign a corresponding grade to the findings of examination as follows: 0, clinically stable sternum (no detectable motion)—normal; 1, minimally separated sternum (slight increase in motion upon special testing—upper limbs, trunk); 2, partially separated sternum—regional (moderate increase in motion upon special testing); and 3, completely separated sternum—entire length (marked increase in motion upon special testing) [14].

#### 2.2.2. Numerical Rating Scale (NRS)

The patient’s pain assessment was conducted via a self-reported NRS of 11 points, where 0 means no pain and 10 means worst pain [28]. The NRS can be verbally administered without the use of physical materials. NRS showed evidence of acceptability, reliability, and validity [29,30]. NRS values of 1–3 indicate mild pain, 4–6 moderate pain, and ≥7 severe pain [31].

Moreover, the perceived level of effort during postural changes was rated using the NRS (0–10).

### 2.3. Procedure

After recording the subjects’ anthropometric data and evaluating sternal stability with the SIS scale, subjects were asked to perform the postural change from the supine to the sitting position. The postural change was carried out with two distinct techniques: (a) rotational postural change [18] and (b) postural change using IDSS. The IDSS was a simple durable strip of inelastic fabric, soft to the touch, washable, 390 cm long, and 10 cm wide. Folded in two, it was tied to the foot of the patient’s bed by means of a lark’s head knot (see the Appendix A).

For both techniques, before performing the postural change, patients were in the semi-Fowler’s position, lying on the bed with a trunk inclination of 30° [32], while at the end of the postural change they reached the sitting position on the bed with their feet on the floor. One familiarization trial for each modality was allowed and the instructions were standardized as follows:Rotational postural change [18]: (a) move your feet towards the edge of the bed; (b) roll on the side; (c) lower the legs from the edge of the bed; and (d) reach the sitting position by helping yourself with the arms (pushing to the side), trying to keep them as close as possible to the trunk.Postural change using IDSS: (a) move the legs towards the edge of the bed, putting them diagonally; (b) tighten the IDSS with one hand, turning the palm upwards; (c) with the other hand, always with the palm facing upwards, grab the hand that already holds the IDSS; (d) pull the IDSS by bending the elbows and, at the same time, activate the abdominal muscles, bringing the trunk to 90° with respect to the bed plane; and (e) place your legs completely outside the bed top and settle in a sitting position.

During this procedure, a dynamometer was placed between the IDSS and the footboard of the bed in order to measure the peak force, expressed in kilograms–weight, exerted by the patient on the IDSS.

### 2.4. Follow-Up

At discharge from the cardiac rehabilitation ward, patients were instructed to use the learnt sternal precautions, such as performing the supine-to-sitting postural change with their most pain-free modality. Patients were interviewed through a follow-up call after 6 months from the discharge in order to report if they had serious complications, as would infection or sternal instability, that required other medical interventions.

### 2.5. Statistical Analysis

Mean ± standard deviation (SD) values were used for descriptive statistics and figures.

A test for normality (Shapiro–Wilk) was performed in all recorded variables. Comparisons of mean pain levels during the three postural conditions (at rest, rotational postural change, and IDSS) were made by one-way repeated-measure ANOVA tests. When ANOVA gave a significant result, the post-hoc Tukey test was conducted to assess differences between postural conditions. To detect differences in perceived level of effort between the two postural conditions (rotational postural change and IDSS) a paired Student’s t-tests was performed.

The clinical meaning of differences between the two modalities of postural change was assessed through the calculation of the Cohen’s d effect size, with the commonly used interpretation that refers to effect sizes as small (d = 0.2), medium (d = 0.5), and large (d = 0.8) based on benchmarks suggested by Cohen [33].

Then, to evaluate the effect of each risk factor (sternal instability, smoking history, presence of macromastia (only women), chronic obstructive pulmonary disease (COPD), obesity, and diabetes) on pain, a subgroup analysis was performed with patients divided for the presence/absence of the main factor risk. Sternal instability was defined as a score ≥ 1 on SIS, while the presence of macromastia as an EU bra size > 80. For each risk factor, a 2-way repeated-measures ANOVA was conducted with the groups (with or without each factor risk) as independent factors and within the three postural conditions (at rest, during rotational and IDSS postural change). When ANOVA gave a significant result, the post-hoc Tukey test was conducted to assess differences between postural conditions.

Spearman’s rank correlation was used to evaluate correlation among variables. Correlation coefficients (ρ) were interpreted as follows: strong correlation if the coefficient value lies between 0.50 and 1; moderate if the value lies between 0.30 and 0.49; and fair if the value lies below 0.29 [34].

Sample size was calculated from El-Ansary et al. [35]; in this study, patients who underwent cardiac surgery via median sternotomy reported a mean pain of about 3.1 points (SD = 2.3) during the supine-to-sitting postural change two weeks after surgery. Since a change of 2 points on NRS was found to be clinically meaningful [36], we expected a difference in pain between the rotational postural change and IDSS postural change of about a 0.5 effect size. Therefore, a sample size of at least 84 patients was required to detect differences between the two postural changes with 90% power (alpha = 0.05, 2-tailed).

Statistical analysis was performed using Statistica software (version 7.1, StatSoft Inc., Tulsa, OK, USA).

## 3. Results

Of 181 patients admitted to our cardiac rehabilitation ward with median sternotomy, 92 patients (51%) met the inclusion criteria. The participants’ demographic and clinical data are reported in Table 1.

Patients were admitted and assessed between the fifth and the fifteenth day after surgery, with a mean time of 8.14 ± 2.62 days. The mean traction force exerted on IDSS during the postural change was 10.13 ± 4.20 kg.

On the SIS, 79 patients (86%) reported a score of 0 (stable, no sternal movements), 7 (8%) a score of 1 (minimally separated sternum), 5 (5%) a score of 2 (partially separated sternum), and only 1 (1%) the worst score of 3 (completely separated sternum).

### 3.1. Pain and Perceived Level of Effort

The mean values of pain and perceived level of effort during the three postural conditions are shown in Figure 1. The mean pain on NRS was 0.73 ± 1.44 points at rest, 5.34 ± 2.32 points during the rotational postural change, and 1.04 ± 1.40 points when performing the postural change with the IDSS. Pain was significantly different between the conditions (ANOVA, F(2,91) = 250.70; *p* < 0.0005). Post-hoc analysis showed that pain was significantly higher during the rotational postural change with respect to the other two conditions (*p* < 0.0005). No difference in pain was found between rest and IDSS. The difference of pain during rotational postural change and the other two conditions showed a large effect size (d = 2.2 when compared to rest, d = 2.3 when compared to IDSS). When analyzing the distribution of pain during the three different postural conditions, it emerged that patients experienced severe pain only in the rotational condition (Figure 2).

Patients reported a mean perceived level of effort on NRS of 6.2 ± 2.5 points during the rotational movement and of 1.6 ± 1.7 points when using the IDSS. The difference between the two modalities of postural change was significant (*t*-test, *p* < 0.0005) and large, with an effect size of 1.84.

Subgroup analysis revealed that subjects with sternal instability showed more pain than stable patients (ANOVA, F(1,90) = 4.79, *p* < 0.05), but there was no interaction between pain and conditions. Moreover, female patients with macromastia had higher pain value than those without macromastia (ANOVA, F(1,22) = 7.21, *p* < 0.05), even if there was no interaction between pain and conditions. No differences were found in the other subgroup analyses.

Pain at rest did not correlate with other clinical parameters, such as the time from surgical intervention (ρ = 0.001). A strong correlation was found between pain and perceived level of effort during the rotational postural change (ρ = 0.64, *p* < 0.0005), while a moderate correlation emerged during the IDSS postural change (ρ = 0.42, *p* < 0.0005). No other correlations were found between the remaining clinical variables and the assessment data.

### 3.2. Follow-Up Call

The 100% of the sample went home with the indication to use the IDSS to perform the supine-to-sitting postural change, as 93% of them reported less pain with respect to the rotational movement and the 7% reported equal pain. At the 6-month follow-up, no patients reported wound infection and no patients with a sternal instability at the previous clinical assessment (SIS > 0 points) reported sternal complications that required rehospitalization. Only a patient (1.1% of the whole sample), a woman without sternal instability during the rehabilitation recovery, had a subsequent sternal displacement.

## 4. Discussion

The aim of this study was to assess which modality of supine-to-sitting postural change, through a rotational movement or performed using the IDSS, induced lower levels of pain and perceived effort in post-operative patients with sternotomy. The results show that the postural change performed using the IDSS was less painful compared to the rotational modality. The level of pain reported by patients during the resting position and in performing the postural change using the IDSS was mild and almost similar between the two conditions. On the contrary, patients reported moderate levels of pain in performing the rotational postural change.

The risk of sternal complications following median sternotomy is frequently reported in literature, suggesting that it would be customary to instruct patients to adopt sternal precautions in order to reduce the incidence of complications [37,38]. Nevertheless, only few studies are available in the literature supporting or refuting the usefulness of sternal precautions [19,25,37,38]. Despite this, lack of literature led to adopting precautions that appear to vary from facility-to-facility; these measures generally include a restriction on upper extremity movements and lifting limitations [19].

Although these hindrances appear to be important, they seem not to be sufficient in order to reduce the risk of complications. Recent studies showed that in the same way, transfers and elevating both arms simultaneously overhead creates the greatest sternal separation [35,39]. In particular, small magnitude of multi-planar motion at the sternal edges occurs during both dynamic upper limb and trunk tasks in patients over the first 3 postoperative months post-sternotomy [39]. In addition, Irion et al. [37,38] measured supra-sternal skin movement during a variety of daily activities, finding that the greatest skin movement occurred during supine-to-sitting using upper extremities, while the least movement occurred when lifting containers up to 5 kg of water (approximately 8 lbs). Moreover, patients with chronic sternal instability experienced the greatest amount of pain during transitions from supine to short sitting. In fact, not surprisingly, in our study, the higher levels of pain (7–10 on NRS) were reported by patients only during the rotational postural change, which is the most common way of performing the supine-to-sitting movement.

Therefore, addressing limitations of the supine-to-sitting transfer seems to be important following median sternotomy. Although modifications to supine-to-sitting postural change using the technique encouraged as part of sternal precautions were shown to decrease stress compared to a typically discouraged method [37,38], these modifications still produced more stress than lifting 5 kg, which is the maximum load allowed to raise for patients [37,38]. As a consequence, since modifications proposed until now in the literature to the supine-to-sitting transfer do not seem sufficient, the results concerning pain of our study may be of considerable importance in the clinical practice for the management of patients with sternotomy.

In fact, finding alternative solutions appears to be essential for guaranteeing the safety of patients. Our work confirmed that more than 50% of patients who undergo sternotomy surgery report persistent pain in the sternum. These findings are superimposable to the study of Moore et al. [40], who found that chest incisional pain was reported by 60% of subjects 3 weeks after cardiac surgery. However, in comparison to the rotational postural change, the use of IDSS in performing the supine-to-sitting postural change resulted in a reduction in the level of pain in our post-operative patients with sternotomy, associated with a decrease in perceived level of effort. Although we never directly measured the forces acting on the sternotomy, we believe that the reduction in the levels of pain and perceived level of effort using the IDSS may indirectly reflect a reduction in the forces acting to the sternum. Indeed, patients used a traction force of about 10 kg for IDSS postural change, which may be comparable to that commonly produced by opening a door, and which was well below the forces occurring during a sneeze or cough [25].

Moreover, the use of IDSS seems especially important in the elderly who are accustomed to carrying out the supine-to-sitting transfer using the rotational pattern compared to younger adults who preferred to perform the postural change through the long sitting position [41]. This behaviour could be due to the fact that older individuals may have decreased abdominal strength and postural stability with an increased fear of falling, and the rotational strategy allowed them to keep an elbow on the mat and maintain a larger base of support [42]. Since patients are often already accustomed in daily pre-surgery for carrying out the rotational movement, they may instinctively continue with the same modality even after the surgical operation, despite the fact that it makes them feel pain.

Among risk factors associated with mechanical forces acting upon the sternotomy site, some were identified also in our sample: chronic obstructive lung disease [17,43], macromastia [8,44], and obesity [16,17,45]. However, maybe due to the small sample, we found higher perceived levels of pain and effort only in patients with macromastia. This supports the hypothesis that women with larger breasts are subjected to increased inferolateral tension across their sternotomy [46], with higher perception of pain caused by the stress force. On the other hand, the presence of diabetes and smoking history did not influence the perception of pain and effort.

Indeed, patient-reported pain was cited as being the main restriction used to guide exercise prescription and progression in the return to daily living movements [47]. In the present study, no patient reported levels of pain and/or perceived effort that were higher in using IDSS compared to performing the rotational postural change. By using IDSS in the correct way, forces exercised by the upper limbs and the abdominal muscles acted symmetrically on the chest, avoiding those asymmetrical forces that are produced during the rotational postural change. As a matter of fact, in reaching the sitting position from the lateral decubitus, the upper limbs produce thrust forces with different intensities and directions, which may impact asymmetrically on the ribcage, placing under stress the sternal wound [39,48].

The good correlation found between pain and perceived level of effort, in both the modalities of postural change, may suggest that weaker patients experienced a higher level of effort in the strain, and consequently, more pain. However, this hypothesis needs to be verified.

Finally, only the 1.1% of the total sample reported a complication, i.e., sternal instability, 6 months after hospital discharge. This result is in line with the noted cases of complications in literature (1–8% [14]). Thus, it is possible to suppose that the precautions given at the discharge, such as the use of IDSS, were opportune for the safety of the patients.

### Limitations

Results of our study are based only on patient-reported outcome measures (pain and perceived effort). One of the limitations was the lack of other types of measure, such as ultrasound. This useful technique might be use in future research to provide valuable real-time feedback regarding sternal healing in patients following cardiac surgery via median sternotomy and may allow for quantitatively assessing motion at the sternal edges. Another limitation was the simple follow-up call at 6 months after hospital discharge; since it is known that sternal healing continues beyond the first three post-operative months [49], we might expect that using a monthly follow-up could allow to find the differences in pain during the two postural steps. Moreover, the heterogeneous but limited sample size did not allow for obtaining solid conclusions about differences in pain and perceived effort in each subgroup of patients stratified by risk factors.

Future research should implement the actual results measuring the motion at the sternal edges during the supine-to-sitting postural change, following patients over a post-operative period of at least 3 months, and assessing the supine-to-sitting abdominal muscles activation and the use of upper limbs to verify the relationship existing between pain, perceived effort, and level of effort in the strain. Moreover, future studies might stratify the population with sternotomy by clinical characteristics in order to give appropriate instructions for each subgroup.

## 5. Conclusions

Sternal displacement is associated with unilateral and rotational movements and the supine-to-sitting postural change is one of the most challenging because of the force applied to the sternum. Thus, since the use of the IDSS seems to reduce the levels of pain and perceived effort during the supine-to-sitting postural change, future studies with quantitative assessments are required to verify its effectiveness and to suggest the adoption of this technique, mostly in patients with high levels of pain and in those with sternal instability.

## Figures and Tables

**Figure 1 jcm-12-04665-f001:**
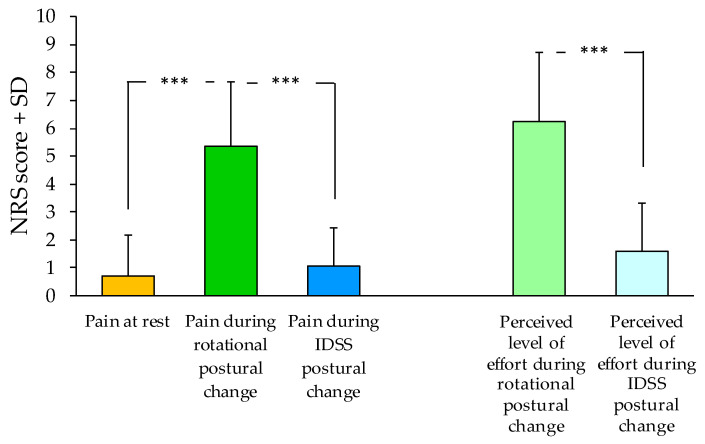
Comparison between pain at rest and across both postural changes, and between perceived level of effort during rotational and IDSS postural change; *** *p* < 0.0005.

**Figure 2 jcm-12-04665-f002:**
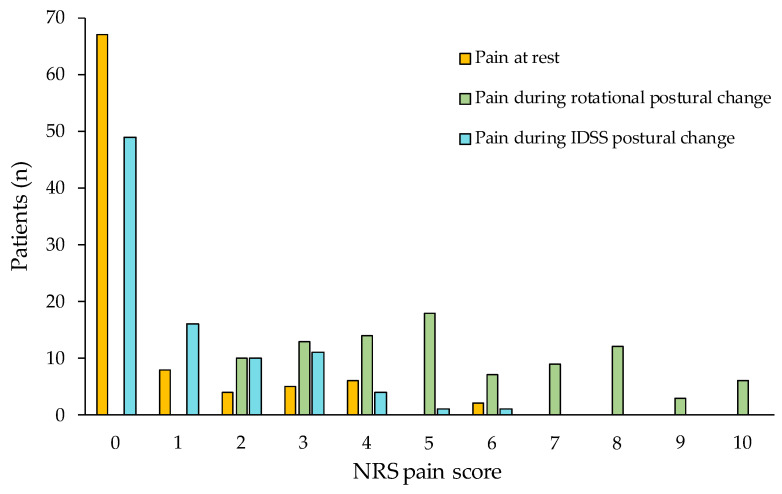
Frequency of pain distribution, assessed by NRS, at rest, during rotational and IDSS postural changes in the whole sample (*n* = 92).

**Table 1 jcm-12-04665-t001:** Characteristics of enrolled patients (*n* = 92).

	Mean	SD
Age (years)	66.37	12.05
Sex (M/F)	68/24	
Height (cm)	167.14	8.31
Weight (kg)	73.66	13.35
BMI index	26.25	3.52
Obesity (yes, no)	13/79	
Thorax size (cm)	99.70	8.29
Macromastia (yes/no, only women)	6/18	
Diabetes (yes/no)	16/76	
COPD (yes/no)	2/90	
Smoking history (yes/no)	22/70	
Time from surgery (days)	8.14	2.62

Abbreviations: BMI, body mass index; COPD, chronic obstructive pulmonary disease; F, female; and M, male.

## Data Availability

The data presented in this study are available on request from the corresponding author. The data are not publicly available due to privacy restrictions.

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
