# Peer review of "Comparison of Two Techniques Performing the Supine-to-Sitting Postural Change in Patients with Sternotomy"

_jcm, 2023, doi:10.3390/jcm12144665_

Round 1

Reviewer 1 Report

In the present study the authors compare two techniques for performing the supine to sitting postural change in patients with sternotomy, evaluating 92 patients for level of pain and for perceived effort.

Since the use of individual device for supine-to-sitting (IDSS) reduced the levels of pain and perceidev effort, the authors concluded suggesting to adopt this technique.  

This is a very interesting work mainly because The topic addressed is relevant for clinical practice and frequent in the daily practise of the post-surgical phase.

The paper is well done and well written. The aims are clearly stated. The methods are to some extent adequate to test the aims.

However, I have major concerns about this paper:

-          even if this point is mentioned in “limitation of the study” section, the number of evaluated patients (92 total but only 24 females, 13 not-stable sternum, 16 with diabetes, 2 with COPD) is too low to obtain solid conclusions and subgroups analysis is rather impossible to be considered

-          the pain and effort evaluations are based on “qualitative” scale, while collecting more objective data could better support the results

-          all the subjects were evaluated only the day of admission to the Cardiac Rehabilitation without a follow up to better support the suggestion to adopt IDSS in the daily practice. 

I strongly believe that the study is interesting, but more cases, quantitative evaluations and a follow-up are needed to draw any reliable conclusion.

Moderate edit of Engish language required

Author Response

Response to Reviewer 1 Comments

In the present study the authors compare two techniques for performing the supine to sitting postural change in patients with sternotomy, evaluating 92 patients for level of pain and for perceived effort.

Since the use of individual device for supine-to-sitting (IDSS) reduced the levels of pain and perceidev effort, the authors concluded suggesting to adopt this technique. 

This is a very interesting work mainly because. The topic addressed is relevant for clinical practice and frequent in the daily practise of the post-surgical phase.

The paper is well done and well written. The aims are clearly stated. The methods are to some extent adequate to test the aims.

However, I have major concerns about this paper:

-          even if this point is mentioned in “limitation of the study” section, the number of evaluated patients (92 total but only 24 females, 13 not-stable sternum, 16 with diabetes, 2 with COPD) is too low to obtain solid conclusions and subgroups analysis is rather impossible to be considered.

Thank you for your suggestions. We agree that the data of this study cannot lead to a definitive conclusion on the use of the IDSS in patients with sternotomy and, in more extent, in subgroups of patients. In this view, throughout the text we have tried to mitigate claims about the importance of the use of the IDSS. We recognized that our results regarding the reduction of the levels of pain and perceived effort, though important from patient’s prospective, are not enough for drawing such strong conclusion about the adoption of this technique.

Regarding the subgroup analysis, we believe that the sample collected in our study is quite in line with the literature and it is representative of the population of patients with sternotomy encountered in clinical practice. In fact, female patients are less than male (31% vs 69% in Zacharias & Habib; 30% vs 70% in Heilmann et al.; 26% vs 76% in our sample), the incidence of post-surgical complications such as sternal instability is estimated to be up to the 8% of the total sample (El-Ansary, et al., 2019), diabetes is presented in about the 9% of patients with sternotomy (Zacharias & Habib, 1996) and patients with COPD were less than 5% (Minami, et al. 2021).

Nevertheless, we have taken into account your suggestion about increasing the recruited sample in order to improve the subgroups analysis. However, we believe that, in this specific study, this choice would not be accurate due to several reasons: 1) first, the total sample size, that is 92 patients with sternotomy, was accurately calculated in order to analyse our primary aim (pain reduction); 2) secondly, we might suppose that enrolling patients with the same procedure expected by our methods, that is enrolling those consequentially admitted to the Cardiac rehabilitation ward with sternotomy and who met the inclusion criteria, would result in the same representative population (such as, less women, less COPD, etc); 3) on the contrary, choosing patients with specific clinical characteristics (i.e. women, patients with diabetes, etc) would be a selection bias. Certainly, for providing an accurate analysis of subgroups, in future studies the sample size should be calculated specifically for each subgroup.

However, we have added your consideration about findings related to the subgroups in the limitation section (page 9, lines 351-354 and 359-361).

El-Ansary, D., LaPier, T. K., Adams, J., Gach, R., Triano, S., Katijjahbe, M. A., ... & Cahalin, L. P. (2019). An evidence-based perspective on movement and activity following median sternotomy. Physical therapy, 99(12), 1587-1601.

Heilmann, C., Stahl, R., Schneider, C., Sukhodolya, T., Siepe, M., Olschewski, M., & Beyersdorf, F. (2013). Wound complications after median sternotomy: a single-centre study. Interactive cardiovascular and thoracic surgery, 16(5), 643-648.

Minami, K., Kabata, D., Kakuta, T., Fukushima, S., Fujita, T., Yoshitani, K., & Ohnishi, Y. (2021). Association between sternotomy versus thoracotomy and the prevalence and severity of chronic postsurgical pain after mitral valve repair: an observational cohort study. Journal of Cardiothoracic and Vascular Anesthesia, 35(10), 2937-2944.

Zacharias, A., & Habib, R. H. (1996). Factors predisposing to median sternotomy complications: deep vs superficial infection. Chest, 110(5), 1173-1178.

-          the pain and effort evaluations are based on “qualitative” scale, while collecting more objective data could better support the results

As previously stated, we agree that qualitative measures are not enough for drawing definitive conclusions, while objective measures - such as sternal micromovements - are fundamental and needed to be investigated before introducing the IDSS into clinical practice.

In a small number of patients (20 out of the 92 recruited), we have already tried to measure the sternal movement, specifically at the middle of the sternum, with the use of an ultrasound. Please see the table attached.

These results appear to be similar to those already reported in the literature, where it has been already found that cough is a condition that consistently increases the displacement of the sternum. Moreover, we found no differences between the two type of supine-to-sitting postural change (t test, p=0.65). This result is quite different from what we expected, since in our manuscript we reported a different patient’s perceived level of pain between the two modalities of postural change. Interestingly, these preliminary findings support our hypothesis about the importance of assessing pain, even if it is recognized as a “simple” qualitative measure. In fact, in clinical practice, pain is the main measure for the clinical choices that physiotherapists must face when approaching patients with sternotomy, all the more since the relationship between sternum displacement and pain has not been well defined yet (El-Ansary, et al., 2018). In particular, bilateral upper limb movements, such as the supine-to-sitting postural change proposed in this study, are usually significantly less associated with sternal pain than other type of movements (El-Ansary et al., 2007).

However, we have decided to do not include these measures for two main reasons: 1) only a small sample of patients underwent this evaluation; 2) the measures were taken only in the middle of the sternum, while it is known that movement at the mid-sternum may vary from motion at the manubrium/upper sternum and the lower sternum (Balachandran et al., 2019). In fact, these preliminary assessments were performed as “practical trials” with the purpose of training physiotherapists in performing ultrasound imaging of the sternum; this training was performed by a senior radiologist in the view of performing a future study.

In fact, in line with your considerations, we aim to conduct a future work in which we will evaluate data about the displacement of the sternum in different conditions and investigate the relationship between sternal pain and objective sternal motion in different intercostal space (mid-sternum).

Balachandran, S., Denehy, L., Lee, A., Royse, C., Royse, A., & El-Ansary, D. (2019). Motion at the sternal edges during upper limb and trunk tasks in-vivo as measured by real-time ultrasound following cardiac surgery: a three-month prospective, observational study. Heart, Lung and Circulation, 28(8), 1283-1291.

El-Ansary, D., Waddington, G., & Adams, R. (2007). Relationship between pain and upper limb movement in patients with chronic sternal instability following cardiac surgery. Physiotherapy theory and practice, 23(5), 273-280.

El-Ansary, D., Waddington, G., Denehy, L., McManus, M., Fuller, L., Katijjahbe, M. A., & Adams, R. (2018). Physical assessment of sternal stability following a median sternotomy for cardiac surgery: validity and reliability of the sternal instability scale (SIS). Int J Phys Ther Rehab, 4(140), 2.

Huang, A. P. S., & Sakata, R. K. (2016). Pain after sternotomy-review. Revista Brasileira de Anestesiologia, 66, 395-401.

-          all the subjects were evaluated only the day of admission to the Cardiac Rehabilitation without a follow up to better support the suggestion to adopt IDSS in the daily practice.

Thank you for this suggestion. Unfortunately, we didn’t ask subjects to return to the facility to be followed-up with the same assessments performed during hospitalization. However, we performed a phone follow-up, about six months after the discharge from the Cardiac rehabilitation ward.

At discharge, patients were instructed to continue to use the sternal precautions learnt (i.e. to use the IDSS for the supine to sitting postural change); after 6 months, patients were interviewed through a follow-up call in which they were asked if they have had sternal complications or other cardiac events that required other medical interventions. What we found was that one patient, a woman without sternal instability during the rehabilitation recovery, have had a subsequent sternal displacement.

We did not include this follow-up in the first draft of our manuscript since it seemed to be of less importance due to the lack of qualitative or quantitative measures, not having assessed patients at the Institute. However, we now recognize that any additional information may be useful for the reader for understanding the evolution of this population.

Therefore, information and data about this phone follow up are now reported in the manuscript in the methods (page 4, lines 151-156), results (page 7, lines 246-254) and discussion (page 8, lines 340-343) sections.

I strongly believe that the study is interesting, but more cases, quantitative evaluations and a follow-up are needed to draw any reliable conclusion.

We hope that the new version complies with your requests. As we previously stated, we recognize the importance of performing a more extended study but, at the same time, we believe that this study strongly supports our hypothesis that assessing the perceived levels of pain and effort is of primary importance for the clinical practice.

In fact, on the basis of this current study, we are going to perform a new one which will consider not only the patient’s perceived measures, but also the real-time ultrasound assessment of the sternal movement. Sample size still need to be calculated, but we will consider your suggestions about the different populations, maybe considering specific sample size for each subgroup of patients. 

Reviewer 2 Report

In order to determine which postural change modality should be recommended to patients with sternotomy, this observational cross-sectional study poses the PICO question of which modality was less painful and evoked less perceived effort: the rotational or that performed with the use of the IDSS. The structure of the research makes sense. The methodology section is very clear and well-defined. Power 90% sample computation. Good-looking tables and graphs displaying the results. Discussion including a synopsis of the findings, their significance, any caveats, and potential next steps. Summary that relates to the PICO question and the research approach utilized.

Author Response

We would like to kindly thank the Reviewer for the appreciation of our work.

Round 2

Reviewer 1 Report

Dear Authors,

I deeply appreciated your efforts to improve the paper and your enthusiasm about this issue.

Unfortunately, the sample size is crucial to support your message and more patients must be tested for this purpose.

No comment

Author Response

Dear Reviewer,

Thank you for your suggestion. We understand the importance of a good sample size. On the basis of the actual paper, in our future work assessing the sternal displacement during the use of IDSS, we will consider to test a more numerous sample, taking into account more specific characteristics related to our actual subgroups.